# Learnable Fourier Features for Multi-Dimensional Spatial Positional Encoding

**Yang Li**
Google Research
Mountain View, CA
liyang@google.com

**Si Si**
Google Research
Mountain View, CA
sisidaisy@google.com

**Gang Li**
Google Research
Mountain View, CA
leebird@google.com

**Cho-Jui Hsieh**
UCLA
Los Angeles, CA
chohsieh@cs.ucla.edu

**Samy Bengio**[*]
Google Research
Mountain View, CA
bengio@gmail.com

## Abstract

Attentional mechanisms are order-invariant. Positional encoding is a crucial component to allow attention-based deep model architectures such as Transformer to address sequences or images where the position of information matters. In this paper, we propose a novel positional encoding method based on learnable Fourier features. Instead of hard-coding each position as a token or a vector, we represent each position, which can be multi-dimensional, as a trainable encoding based on learnable Fourier feature mapping, modulated with a multi-layer perceptron. The representation is particularly advantageous for a spatial multi-dimensional position, e.g., pixel positions on an image, where $L_2$ distances or more complex positional relationships need to be captured. Our experiments based on several public benchmark tasks show that our learnable Fourier feature representation for multi-dimensional positional encoding outperforms existing methods by both improving the accuracy and allowing faster convergence.

## 1 Introduction

Attentional mechanisms are a central component in many deep architectures [1, 25], which allow a model to selectively focus on specific information in the context. Transformer [38] and its many variants, such as [29, 38, 16, 3], which are solely based on attentional mechanisms, have advanced the state of the art on many tasks that involve data with inherent temporal and spatial orders, e.g., machine translation [38], image generation [16], and object detection [3].

In contrast to recurrent [14, 34, 27] or convolutional architectures [18], which automatically capture the ordinal information as computation progresses based on sequential or spatial dependencies, attentional mechanisms are order invariant. It allows a model to directly access information at an arbitrary position in a sequence or space. The lack of ordinal information in the model is not an issue when attentional mechanisms are combined with a recurrent or convolutional architecture [1, 25]. However, it is crucial for Transformer-alike models where the entire model is built based on attentional mechanisms.

To capture positional information in the data, e.g., the token position in a sentence or the pixel coordinates in an image, *positional encoding* has been introduced [10, 38], where a position in a

---

[*]Currently at Apple.

35th Conference on Neural Information Processing Systems (NeurIPS 2021).

one or two-dimensional space is mapped to a vector space by either learning or heuristics-based approaches. The representation of an input, by combining both its positional encoding and content representation, e.g., word embeddings, then participates in downstream computation for attentional mechanisms. The original Transformer model uses a fixed sinusoidal encoding with predefined wavelengths [38]. However, the predefined features lack flexibility and may not capture important position information in a task-dependent manner. To encode positions in a more flexible and data-driven way, position embedding approaches (e.g., one used in BERT [8]) introduce trainable embedding vectors for each (absolute or relative) position. Unfortunately, this data-driven approach comes at the cost of introducing a large amount of extra learnable parameters proportional to sequence lengths times the hidden dimension size. Moreover, it is non-trivial to apply position embedding to problems with variable sequence lengths.

In this paper, we consider the problem of designing a position encoding for multi-dimensional spatial positions, such as pixel positions in an image or object bounding boxes in a spatial structure such as UIs. Existing methods typically use sinusoidal position encoding with hand-crafted frequencies or learned embedding to encode each dimension independently and then combine the resulting vector representations via concatenation, e.g., [29, 3, 9]. Unfortunately, these approaches, by concatenating the representation of each dimension, are not effective to capture desired positional similarity on an image, such as $L_2$ distance or more complex positional relationships. While embedding-based approaches have the potential to learn complex positional relationships, since the number of unique positions grows exponentially to the input dimension, the approach incurs large overhead in 2D and could be infeasible scaling to a higher dimensional space. In addition, special treatments are needed to adjust the learned position embedding when the test image sizes differ from training, such as bicubic interpolation used in DeiT [37] or Vision Transformer [9]. To avoid these special adjustments, it is an important for positional encoding to handle unseen positions.

The main contributions of our work are as follows. We design a novel positional encoding method that learns a function to map multi-dimensional positions into a vector space. The function extracts position information based on a set of Fourier features and passing them to an MLP. The encoding function is *learnable* and is initialized in such a way that the inner products of our positional encodings approximate Euclidean distances. The inductive bias can be desirable in a 2D or higher-dimensional space and by learning from the data, the representation can be adapted to a specific problem. Since our method learns an encoding function instead of embedding vectors for each position, it is naturally *inductive* and can handle test samples with arbitrary length. Our method is *parameter-efficient*, in the sense that the number of parameters do not grow with sequence length. To allow complex positional relationships, our representation is also *composable* by encoding each subset of dimensions, in a multi-dimensional space, using a shared learnable Fourier features. We evaluate our method on a number of tasks where Transformer-based models have been used for problems with multi-dimensional positions, including image generation [16], object detection [3] and image classification [9], which all involve 2D positions (vertical and horizontal) in images. We also evaluate our method on natural language generation in graphical user interfaces, which involve modeling a sparse spatial structure of UI objects on the screen, where each object is characterized by 4-coordinate values (top, left, bottom, and right) [20]. These experiments show that our positional encoding method consistently outperforms existing methods by both improving accuracy and accelerating learning.

## 2 Background

### 2.1 Positional Encoding

In Transformer models, the self-attentional mechanism determines the strength between each pair of items based on the dot product similarity of their vector representations, which are derived from an item's content embedding and positional encoding [38] (Appendix A). Although positional encoding (PE) does not function alone in determining the attention strength, the benefit of having the inductive bias of positional relevance in the PE is evidenced by the success of the sinusoidal positional encoding originally proposed in Transformer [38] (Equation 1).

$$PE(p, 2d) = \sin \frac{p}{10000^{2d/D}}; PE(p, 2d+1) = \cos \frac{p}{10000^{2d/D}} \tag{1}$$

which encodes a scalar position, $p$, using sinusoidal functions with different constant frequencies for each dimension, $d$, of a $D$-dimensional encoding vector. The dot product of this encoding representation naturally captures positional similarity in a 1D sequence in a parameter-free fashion.

The other category of approaches for PE is to treat each position as a discrete token that can then be uniquely represented as a learnable embedding vector [9, 10, 16, 8]. The approach can capture arbitrarily complex relationships between positions by learning from data, but it can be difficult to generalize for positions that are rarely encountered during training. For example, the heatmap map in Figure 1 shows the positional similarity learned by a Transformer model for a machine translation task on En-De WMT32k [38]. Towards the diagonal, i.e., positions that are closer, there tends to be higher similarity because each token attends to itself the most. However, the trend is diffused for large positions, i.e., when a sequence is long, because fewer training examples have long sequences. For what is followed, a model will not be able to correctly represent large positions in a long sequence at training and test time.

There has been extensive work in extending positional encoding for different modeling tasks, e.g., handling long-range sequences [7, 43] or tree structures [35, 41], or enhancing vision tasks using input-dependent positional encoding [6]. Our work is related to the effort of using a continuous function instead of embedding retrieval for modeling positions. Previous work [40] uses complex embedding functions to model 1D positions. It has been shown that position encoding in a 1D space can be learned as a Neural ODE system [23]. However, their approach cannot be extended to 2D or higher-dimensional problems. More recently, previous work has proposed learnable sinusoidal representations for 1D positions [39] in language tasks. In contrast, we focus on representing 2D or even higher dimensional positions in spatial tasks.

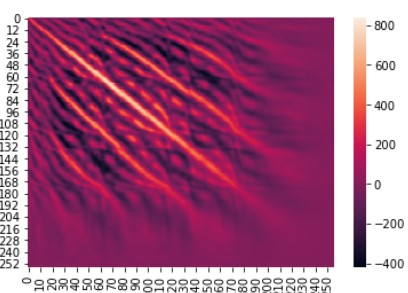

Figure 1: The heatmap shows the dot product similarity of positional embeddings learned by a Transformer model for the En-De WMT32k machine translation task.

Our work is different from the body of work on relative positional encoding, which directly represents pairwise positional relation between query and key [33, 15, 2, 32, 11]. Because there are $O(N^2)$ of pairwise relations for $N$ positions, relative positional attention is only feasible for a small range, e.g., within a clip distance or local range, although recent work [24] has achieved linear complexity by approximating relative positional encoding. Because relative positional encoding directly participates in the computation of the attention matrix, instead of addressing the representation of individual input items, it cannot be easily plugged into many existing Transformer architectures. In contrast, our method is fully compatible with many Transformer benchmarks models. In our work, we focus on representing individual multi-dimensional spatial positions such that these representations achieve desirable pairwise relation later during attention computation.

## 2.2 Encoding Multi-Dimensional Spatial Positions

A common approach for positional encoding for a 2D problem is to encode each positional dimension (vertical and horizontal) independently using either sinusoidal (Equation 1) or direct embedding-based methods, and then concatenate these representations to form the final positional encoding [29, 3, 16, 9]. Although the approach of sinusoidal concatenation allows the model to capture the positional (spatial) relationships orthogonally along each axis, the similarity decays much faster along other directions, as shown in Figure 2(a), which ideally should decay at the same rate along all the directions for modeling $L_2$ distances as shown in Figure 2(b).

While concatenating learned embedding has the capacity to model complex spatial relations between positions, they can be difficult to generalize. It is even brittle for addressing problems involving higher-dimensional positions. For example, for modeling spatial structures in UIs [19, 20], recent work takes a collection of UI objects as input and the positional attribute of each object is its spatial configuration on the screen, which involves 4 coordinate values: [top, left, bottom, right]. The occurrence of unique object positions can be sparse, which makes it difficult for a model to

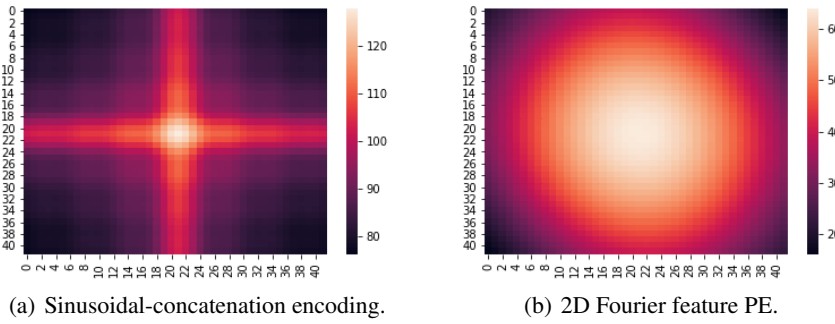

| (a) Sinusoidal-concatenation encoding. | (b) 2D Fourier feature PE. |

Figure 2: The similarities of the center position to the rest positions on the 2D space, based on the dot product between their positional encoding of each approach.

generalize. There are positions that are rarely seen during training might occur at test time. Motivated by these analyses, we intend to develop a positional encoding method for representing a multi-dimensional position by taking into account all the dimensions holistically, and meanwhile enabling effective inductive bias and learnability in the representation.

## 2.3 Fourier Features

The task of mapping data points to a vector space such as their dot product achieves certain distance metric has been extensively investigated in the literature of kernel functions [30, 31, 17, 42, 12, 36].

$$k(x, y) \approx z(x)'z(y)$$

where $x, y \in \mathcal{R}^d$ and $k(x, y)$ is a shift-invariant kernel function; and $z(x)$ and $z(y)$ are feature mapping respectively.

Fourier features [30, 31] are a common technique to approximate a Gaussian kernel, a shift-invariant kernel, with $k(x, y) = \exp(-\frac{\|x-y\|^2}{\gamma^2})$ where $\|x - y\|^2$ is the Euclidean distance between two points, $x$ and $y$, which each point is a multi-dimensional position in our context. This unique attribute inspired us to represent a multi-dimensional position via Fourier features, which is a basis for our approach for positional encoding.

Random Fourier features have also been applied in deep learning models, e.g., approximating the attention matrix in Transformer [5]. Recently, adaptive random Fourier features [21] have been proposed for better kernel approximation that show improvement on classification tasks. In contrast, we propose learnable Fourier features for spatial positional encoding and integrate the method in various Transformer-based deep architectures that show improvements on multi-dimensional spatial tasks.

## 3 Learnable Fourier Features Positional Encoding

We propose to learn a position encoding function that maps an $M$-dimensional position $x \in \mathcal{R}^M$ into a $K$-dimensional feature vector. This $K$-dimensional vector will then be used in downstream computation for attention mechanisms. The proposed encoding function is composed with the following two components:

**Learnable Fourier Features** To extract useful features from the input position $x$, we consider the following feature extraction layer motivated by the idea of Fourier features [30, 31]. Given an $M$-dimensional position, $x \in \mathcal{R}^M$, we acquire a $D$-dimensional Fourier feature vector representation for the position, $r_x \in \mathcal{R}^D$, as follows:

$$r_x = \frac{1}{\sqrt{D}}[\cos xW_r^T \| \sin xW_r^T] \tag{2}$$

where $\|$ is the concatenation of two vectors. This can also be viewed as the generalization of sinusoidal position encoding to the multi-dimensional case, while we set $W_r \in \mathcal{R}^{\frac{D}{2} \times M}$, which

defines both the orientation and wavelength of Fourier features, as trainable parameters. Since $\cos(a - b) = \cos a \cos b + \sin a \sin b$, we have the following:

$$r_x \cdot r_y = \frac{1}{D}\text{sum}\big(\cos((x - y)W_r^T)\big) := h_{W_r}(x - y) \tag{3}$$

where $\cdot$ is the dot product. Therefore, vectors in the form of (2) enjoys the shift-invariance property—the dot product of $r_x$ and $r_y$ is a function of $x - y$ and the function is parameterized by $W_r$. Learning $W_r$ is equivalent to obtaining the most informative function on $x - y$ that can be useful for the downstream task.

In our algorithm, the linear projection $W_r$ is initialized by drawing from a normal distribution

$$W_r \sim \mathcal{N}(0,\, \gamma^{-2}). \tag{4}$$

When the linear projection weights are drawn in such a way, according to random Fourier features [30, 31], the dot product between two feature vectors, $r_x$ and $r_y$, approximates the Gaussian kernel over the original positions.

$$r_x \cdot r_y \approx \exp(-\frac{\|x - y\|^2}{\gamma^2}). \tag{5}$$

Figure 2(b) visualizes this representation, which introduces a useful inductive bias of $L_2$ distances into the model.

**MLP layer**    To feed the representation to the downstream computation, we give the representation additional capacity by modulating the features with a multi-layer perceptron:

$$PE_x = \phi(r_x, \theta)W_p, \tag{6}$$

where $\phi(\cdot)$ is the perceptron parameterized by $\theta$. $W_p$ are trainable parameters for projecting the representation onto a target dimension of positional encoding for combining with content embedding. Our purpose with MLP here is very different from previous work that uses non-linear transformation such as an RNN to capture positional dynamics [26, 23]. These previous works do not handle non-sequential multi-dimensional positions.

The learnable parameters in our position encoding function are $W_r$ for Fourier features and $\theta, W_p$ for the MLP layer. The size of these matrices are independent of the sequence length. Furthermore, the position encoding function can be applied to any input position $x$, so our method can be easily applied when training and testing involve different positions, e.g., images with different resolutions. Compared to the previous sinusoidal representation (Equation 1), our representation is learnable and multi-dimensional. Compared to the discrete embedding-based approach, our representation treats each dimension of a position as a continuous-valued vector, which alleviates the sparsity issue with discrete positions. Previous work has revealed that using sinusoidal activation functions might suffer optimization problems due to vanishing gradients in extreme cases [28], although we do not observe much difficulty in training our positional encodings.

Our representation is applicable for many 2D spatial tasks, e.g., image-related tasks. For tasks involving higher-dimensional positions, the positional similarity between positions might be more complicated than $L_2$ distances. For example, to model the spatial structure of a natural scene or a graphical user interface, given two objects in the structure, $x$ and $y$, coordinate values, $[x_1, x_2, x_3, x_4]$ and $[y_1, y_2, y_3, y_4]$, represent the object's top, left, bottom, and right position. The $L_2$ distance between the two positions $\sum_{i=1}^{4}(x_i - y_i)^2$ will capture neither the minimum nor the maximum distance between the two objects, or any vertical or horizontal alignments of them. To address this issue, we hypothesize that complex spatial relationships can be built on top of shift-invariant relations enabled by our positional encoding. Specifically, we can partition a multi-dimensional position into groups, and apply the same encoding pipeline to each group of coordinate values. The process is similar to applying convolution over partitions with the kernel and stride sizes to be the group size. We can then concatenate the output of all the groups to form the final positional encoding. We will elaborate on this use case in the UI modeling experiment (Section 4.4). An implementation of our positional encoder based on tensor operation is detailed in Algorithm 1 in which Equation 2 and 6 are realized in Line 1 and 2.

---

**Algorithm 1:** Compute the Fourier feature positional encoding of a multi-dimensional position.

---

**Input:** A tensor $X$ in the shape of $[N, G, M]$ that represents $N$ positions where each position is in the shape of $[G, M]$ that represents $G$ positional groups and each group has $M$-dimensional positional values.

**Output:** $PE_X$ in the shape of $[N, D]$ where $D$ is the depth of the positional encoding.

**Hyperparameter**: The depth of the Fourier feature dimension $|F|$, the hidden layer dimension $|H|$, and the positional encoding dimension $D$, and $\gamma$.

**Initialization**: Initialize learnable weights $W_r \in \mathcal{R}^{\frac{|F|}{2} \times M}$ by sampling from $\mathcal{N}(0, \gamma^{-2})$; Initialize learnable weights $W_1 \in \mathcal{R}^{|F| \times |H|}$, $B_1 \in \mathcal{R}^{|H|}$, $W_2 \in \mathcal{R}^{|H| \times \frac{D}{G}}$ and $B_2 \in \mathcal{R}^{\frac{D}{G}}$.

---

**1** $F \leftarrow \frac{1}{\sqrt{|F|}}[\cos X W_r^T; \sin X W_r^T]$ (Eq. 2);

**2** $Y \leftarrow \text{GeLU}(F W_1 + B_1) W_2 + B_2$ (Eq. 6) ;

**3** $PE_X \leftarrow$ Reshape $Y$ into the shape of $[N, D]$;

**4 return** $PE_X$.

---

## 4    Experiments

We evaluate our approach on a range of benchmark tasks using Transformer-based models in comparison with several existing positional encoding methods.

### 4.1    Image Generation

We compare our method with existing positional encoding approaches based on Reformer [16] for the image generation task on the ImageNet 64x64 dataset [4]. Reformer is a Transformer-based model that uses locality-sensitive hashing and reversible residual layers to efficiently handle long sequences. Reformer flattens a 64x64 image into a sequence (Length=64x64x3=12,288) in a raster scan red-green-blue order. Reformer as an auto-regressive model predicts the pixel value at each position by attending to previous positions. We equip Reformer with different positional encoding methods.

- *Embed-2D*: Reformer's default positional encoding concatenates the embedding of each dimension from two embedding matrices: vertical $[64, 384]$ and horizontal $[64, 384]$.

- *Embed-1D*: The baseline method assigns a learnable embedding to each position in the flattened sequence, from an embedding matrix of $[64 \times 64, 768]$, which ignores the 2D structure of an image and lets the model learn positional relations all by itself.

- *Sine-2D* and *Sine-1D*: Similar to Embed-2D and Embed-1D, but they instead encode a position using Transformer's constant sinusoidal formulation (Equation 1).

- *Learnable-Fourier + MLP*: Our method that implements Algorithm 1 using the hyperparameter $|F| = 384$, $|H| = 32$, $D = 768$. We picked these dimensions for our method to have roughly the same number of parameters as Embed-2D, the benchmark of Reformer.

We leave the RGB axis to use the direct embedding as the original Reformer: $[3, 256]$. The concatenation of the pixel position encoding and the RGB index embedding results in an representation that has the same depth (1024) as the one in the original paper, which allows the rest of the model intact.

We follow the experimental procedure as detailed in the Reformer paper. All our experiments used a 6-layer, 8-head-attention Reformer, with $d_{model} = 1024$, $d_{ff} = 4096$, and $n_{heads} = 8$. These models are implemented based on the Reformer codebase in Trax[2]. The training for each Reformer model is parallelized across 32 TPU v2 cores, and each batch contains 8 sequences (images) on each core. We trained each model variant for 100k steps, which took about 24 hours to complete.

As shown in Figure 3a, our method, Learnable-Fourier + MLP, outperforms all the baselines in terms of convergence speed and achieves better accuracy, i.e., lower bits per dim at the end. The Reformer's original positional encoder, Embed-2D, is the second best. Sine-2D clearly outperforms Sine-1D, and Embed-1D achieves a similar performance as Sine-1D.

---

[2]`https://github.com/google/trax/tree/master/trax/models/reformer`

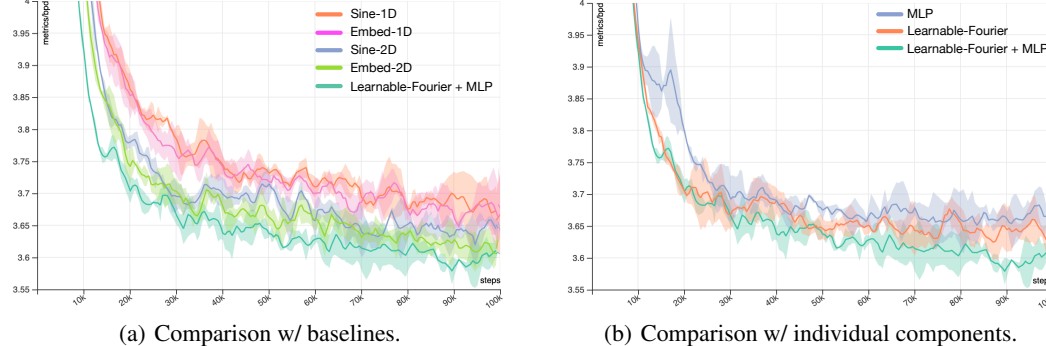

| (a) Comparison w/ baselines. | (b) Comparison w/ individual components. |

Figure 3: Bits per dim (bpd) w.r.t. training steps on evaluating Reformer on the held-out data of the ImageNet 64x64 dataset for image generation, using different positional encoding methods. The plot shows the mean and 95% confidence interval based on 3 repeats of experiments for each method.

To understand how each component in our method contributes to the overall performance, we compare Learnable-Fourier+MLP with its components Learnable-Fourier and MLP alone. MLP takes a 2D position as input and outputs a 768-dimensional positional encoding. Our experiment shows that Learnable-Fourier or MLP alone does not perform as good as their combination, Learnable-Fourier+MLP (see Figure 3b). It is worth noting that Learnable-Fourier shows competitive performance for the first 30k steps, which indicates that it benefits from an effective bias for capturing meaningful positional relationships.

## 4.2 Object Detection

We evaluate the proposed positional encoding in DETR [3], a recent model that uses a Transformer for end-to-end object detection. It uses a Transformer to take the output from a ResNet, i.e., a feature map with the spatial dimensions of $42 \times 42$. Similar to Reformer, positional encoding represents each position in the grid as part of the input to the Transformer encoder in DETR. We experiment with the default 6-layer Encoder-Decoder setup in DETR, with the same set of hyperparameters, on the COCO 2017 object detection dataset [22] that has 118k images for training and 5k for validation. We equip the DETR model with different positional encoding methods, including Sine-2D that is DETR's default method, Learnable-Fourier+MLP, Embed-2D and MLP. The model implementations are based on the DETR codebase[3], which are ported into JAX[4]. The training for each DETR model is parallelized across 64 TPU v3 cores with a batch size of 64 images. We let each model train for 300 epochs to converge, which took about 3 days. We follow the experimental procedure of the DETR paper, and report accuracy on the validation set.

DETR uses image augmentation in both training and validation. Each image is randomly resized to several specific dimensions with the smaller side of the image at one of the following sizes: 480, 512, 544, 576, 608, 640, 672, 704, 736, 768, and 800. For positional encoding, all image positions are normalized to a range of $(0, 1)$. Normalization is valuable because of random resizing and cropping during image augmentation results in images with different sizes. Embed-2D treats each position as a discrete value, and all the methods except Embed-2D leverages position normalization. As shown in Table 1, Learnable-Fourier+MLP offers the best performance across all the metrics. Sine-2D and MLP perform competitively while Embed-2D has the worst performance.

To investigate how each encoding generalizes to unseen image sizes, we modify the benchmark by reserving the three largest sizes: 736, 768, and 800 for validation only. We also disable position normalization. As a result, there are large positions that are never seen during training, which requires each method to generalize (or extrapolate) to these positions. As shown in Table 2, the benefit of Learnable-Fourier+MLP is more pronounced, and the performance gap between Embed-2D and the other methods is further increased.

---

[3]https://github.com/facebookresearch/detr/blob/master/models
[4]https://github.com/google/jax

Table 1: The impact of different positional encodings on DETR for object detection.

| Method | $AP$ | $AP_{50}$ | $AP_{75}$ | $AP_{small}$ | $AP_{medium}$ | $AP_{large}$ |
|---|---|---|---|---|---|---|
| Sine-2D | 40.1 | 60.4 | 42.6 | 18.5 | 43.6 | 58.8 |
| Embed-2D | 39.3 | 59.8 | 41.4 | 18.7 | 42.5 | 57.5 |
| MLP | 40.0 | 60.3 | 42.2 | 18.6 | 43.7 | 58.1 |
| Learnable-Fourier+MLP | **40.2** | **60.7** | **42.7** | **18.8** | **43.8** | **59.1** |

Table 2: Performance of each method for object detection involving unseen image dimensions.

| Method | $AP$ | $AP_{50}$ | $AP_{75}$ | $AP_{small}$ | $AP_{medium}$ | $AP_{large}$ |
|---|---|---|---|---|---|---|
| Sine-2D | 38.9 | 59.6 | 40.9 | 17.5 | 42.5 | 57.5 |
| Embed-2D | 36.6 | 58.2 | 37.7 | 15.9 | 40.0 | 55.3 |
| MLP | 38.6 | 59.5 | 40.3 | 17.1 | 42.1 | 57.1 |
| Learnable-Fourier+MLP | **39.5** | **60.0** | **41.6** | **18.9** | **43.0** | **58.0** |

## 4.3 Image Classification

We evaluate the proposed positional encoding on image classification, another popular task on images, based on Vision Transformer (ViT) [9], a Transformer-only architecture that does not use CNN for image embedding. The default positional encoding in ViT is Embed-1D. In this experiment, we focus on the ViT-B/16 model that is a 12-layer Transformer encoder with Hidden_size=768, MLP_size=3072 and 12 attention heads. The input to ViT-B/16 uses a $14 \times 14$ image grid where each cell corresponds to a $16 \times 16$ image patch. We train each model on the ImageNet dataset for 90 epochs, and report its accuracy on the ImageNet validation dataset. Learnable Fourier+MLP achieved better performance (Precision@1=74.5%) on the validation dataset than Embed1D (Precision@1=73.6%).

Dosovitskiy et.al. investigated several positional encoding methods in their work (see Table 8 in [9]), including Embed-1D, Embed-2D and relative positional encoding. They pre-trained these models on the large JFT dataset (300 million examples) and then report their performance on ImageNet 5-shot linear tasks. They found the model suffers when no positional encoding is used but there are no significant impacts for using each of these positional encoding methods. We suspect that given such a large model (86M Params), it is not difficult for any of these positional encoding methods to learn the small number of unique positions ($14 \times 14 = 196$) on the image. In their experiment, Embed-1D achieves 64.206% accuracy, Embed-2D 64.001% and Relative Positional Encoding 64.032%. We experimented Learnable-Fourier+MLP in this experiment, which achieved 64.732% accuracy.

## 4.4 Widget Captioning

So far, we have investigated tasks that handle 2D positions in an image. In this experiment, we investigate even higher-dimensional positions. In a widget captioning task [20], the model is trained to generate natural language description of widgets in graphical user interfaces, e.g., buttons and icons. A significant part of the model is to encode a UI screen structure, which consists of a collection of 2D objects of different sizes, using a Transformer encoder. To represent the spatial configuration of each object, the original model assigns a learnable embedding vector to every discrete coordinate value of each dimension of the object bounding box, including the left, top, right, and bottom dimensions. The four embedding vectors then jointly represent a bounding box on the screen. We refer this baseline as *Embed-4D*. Li et al. found that position encoding has a significant impact on the performance of widget captioning models (see Table 6 in Appendix F of the paper [20]).

Because there is no obvious distance metrics between bounding boxes, we hypothesize that an appropriate metric can be learned on top of $L_2$ distances of specific dimensions. To do so, We evaluate three different partitions of bounding box dimensions, and use our method to encode each group in parallel as detailed in Algorithm 1: Learnable-Fouier+MLP-1/4 treats all the 4 coordinate dimensions [(top, left, bottom, right)] as one group, i.e., $G = 1$; Learnable-Fourier+MLP-2/2 splits the 4 dimensions into 2 groups [(top, left), (bottom, right)], i.e., $G = 2$; and finally Learnable-Fourier+MLP-4/1 encodes 4 groups of 1-dimensional value [(top), (left), (bottom), (right)], i.e., $G = 4$. We also add the sinusoidal approach to the comparison, which

represents each positional dimension separately and then uses their concatenation to as the positional encoding a bounding box (referred as *Sine-4D*).

Table 3: The performance of different positional encoding methods on the widget captioning test set. SOTA shows the results from the original paper, which is reproduced by Embed-4D in our experiment.

| Method | BLEU-1 | BLEU-2 | ROUGE | CIDEr | METOER | SPICE |
|---|---|---|---|---|---|---|
| SOTA [20] | 44.9 | 32.2 | 44.7 | 97.0 | 31.7 | 17.6 |
| Embed-4D | 45.2 | 31.9 | 45.0 | 97.0 | 31.7 | 17.3 |
| MLP | 34.0 | 23.5 | 33.7 | 70.3 | 23.7 | 10.2 |
| Sine-4D | 44.9 | 31.9 | 43.9 | 94.9 | 31.0 | 16.7 |
| Learnable-Fourier-2/2 | 44.9 | 31.6 | 44.3 | 95.3 | 31.6 | 17.7 |
| Fixed-Fourier+MLP-1/4 | 45.0 | 32.1 | 44.2 | 95.4 | 31.2 | 17.1 |
| Fixed-Fourier+MLP-2/2 | 46.1 | 32.5 | 45.8 | 100.2 | 32.5 | 18.4 |
| Fixed-Fourier+MLP-4/1 | 45.5 | 32.1 | 45.1 | 97.2 | 31.7 | 17.6 |
| Learnable-Fourier+MLP-1/4 | 45.6 | 32.7 | 45.2 | 99.1 | 32.2 | 17.1 |
| Learnable-Fourier+MLP-2/2 | 46.1 | 32.7 | 45.9 | 98.0 | **32.6** | **17.9** |
| Learnable-Fourier+MLP-4/1 | **46.8** | **33.4** | **46.1** | **100.7** | 32.4 | 17.8 |

We use the same model architecture and hyperparameters of the strongest model, *Pixel+Local+Context*, as the original paper [20], and built our experiment based on the public codebase of widget captioning[5]. Specifically, the screen encoder uses a 6-layer, 8-head Transformer with a hidden size of 128. We train all the models to 100k steps with Adam optimizer and a scheduled learning rate detailed the original paper. All the models converged within 12 hours using 4 V100 GPU cores.

All the results are acquired by applying each trained model on the test dataset, based on the same set of captioning metrics. As shown in Table 3, our method outperforms the benchmark method Embed-4D (#Params=5.11M) with a large margin even though our method uses fewer parameters (#Params=5.07M), particularly on BLEU-1, BLEU-2, ROUGE and CIDEr, which clearly advanced the state of the art for this task. Interestingly, both Learnable-Fourier+MLP-2/2 and Learnable-Fourier+MLP-4/1 outperform Learnable-Fourier+MLP-1/4, which indicate that more complex distances needed to be modeled in this task than $L_2$ distances. Compared to Embed-4D, T-tests (over 3 runs of each model) show the gain of Learnable-Fourier+MLP 4/1 is statistically significant ($p < 0.05$) on BLEU-1, ROUGE, CIDEr and METOER; Learnable-Fourier + MLP 2/2 achieves significance ($p < 0.05$) on BLEU-1, ROUGE and METOER. For the two champion conditions, i.e., Learnable-Fourier+MLP-4/1 and 2/2, we found on most metrics there is no statistical significance between their performance ($p > 0.05$). Learnable-Fourier+MLP-4/1 outperforms 2/2 only on CIDEr with marginal statistical significance ($p = 0.042$).

We also included a few ablation studies in this experiment. One variant is to fix Fourier features but still include MLP. In this group, i.e., Fixed-Fourier+MLP-*, Fixed-Fourier+MLP-2/2 clearly performs the best across all the metrics. Overall, it seems that Learnable-Fourier+MLP still has advantages over the fixed one on most cases. We then look at Learnable-Fourier but without using MLP. Learnable-Fourier-2/2 seems to perform worse than its counterpart in the other groups on every metric, which indicates that MLP is a crucial component for positional encoding in this task. Lastly, although using MLP alone as the encoding function seems competitive in the object detection task, it performs poorly in this experiment.

## 5 Discussion

One clear trend that emerges from our experiments is that positional encoding methods that treat an image as a flattened sequence (Embed-1D or Sine-1D) do not perform well, even though the model is given a great capacity to learn these positional relations. We also observe that taking a multi-dimensional position holistically often performs better than representing each dimension separately and then concatenating these representations. We found it generally beneficial to use the

---

[5]`https://github.com/google-research/google-research/tree/master/widget_caption`

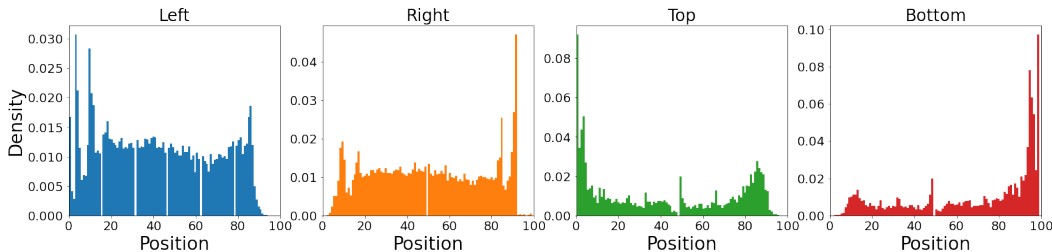

Figure 4: Widget position distributions for each dimension in the training set. There are positions rarely occurred in the training set.

multi-layer perceptron (Equation 6) to process the Fourier features for positional encoding before it is mixed with content embedding. We obtained mixed results for using MLP alone as the positional encoding function, which performs competitively on the object detection task but poorly on the UI modeling task that involves sparse spatial structures. From these experiments, it seems not necessary to use a large random feature dimension to achieve good results.

Table 4: The accuracy of each method on widgets with seen and unseen positions.

| Method | Seen CIDEr | Unseen CIDEr |
|---|---|---|
| Embed-4D | **123.4** | 78.5 |
| Sine-4D | 121.3 | 76.4 |
| Learnable-Fourier+MLP-4/1 | **123.4** | **82.2** |

To understand how different positional encoding methods can generalize to unseen positions, we analyze test results for the widget captioning task. There are positional values rarely or never seen in the training set (Figure 4). Specifically, 1867 widgets in the test set have seen positions and 2692 have unseen positions. Table 4 shows that our method generalizes to unseen positions significantly better than baselines. There are a number of reasons for the proposed positional encoding to generalize for unseen positions. First, it treats positions as continuous-valued vectors. As a result, it does not suffer from the difficulty with embedding-based approaches where an embedding vector is assigned to a discrete position, which can be not trained or significantly under-trained when a position is unseen or rarely seen. Second, the Fourier features capture the relative positional relationships by maintaining the shift-invariant property during learning (Equation 3), which applies to unseen positions as well.

One direction that deserves further investigation is how the interaction between positional encoding and content embedding should be taken into account for the design of a positional encoding function. Our work investigated positional encoding when it is combined with content embedding via addition for all the image tasks. It would be interesting to investigate how our positional encoding performs when it is concatenated with content embedding on these tasks.

Although Euclidean distances might be a desirable positional metric for images, tasks such as widget captioning involves sparse spatial structures, and the spatial relations between two rectangular objects on the screen can be more complicated. For example, similarity between two bounding boxes can be related to their vertical or horizontal alignment or overlaps (IoU), or other domain specific factors related to UI layouts. Our positional encoding method outperformed benchmark methods in this task, which showed that it is better equipped to capture these spatial relationships. Yet, it is worth investigating methods that can more directly capture these complex spatial relationships.

# 6   Conclusion

We present a novel approach for positional encoding based on learnable Fourier features. We evaluate our approach on a range of multi-dimensional spatial tasks, including image generation, object detection, image classification, and sparse spatial structure modeling in user interfaces, which show that our positional encoding consistently outperforms the benchmark methods.

## Acknowledgments

We would like to thank anonymous reviewers for their insightful comments and constructive feedback that have significantly improved the work.

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
