Learnable Fourier Features for Multi-Dimensional Spatial Positional Encoding: Appendix

## A    Attention-Based Models

We review positional encoding in the context of Transformer models [38]. The central building block of these models is multi-head attention and each attention head is calculated as follows:

$$\text{Attention}(Q, K, V) = \text{softmax}(\frac{QK^T}{\sqrt{d_k}})V \tag{7}$$

where queries $Q \in \mathcal{R}^{N \times d_k}$, keys $K \in \mathcal{R}^{N \times d_k}$, and values $V \in \mathcal{R}^{N \times D_v}$. $N$ is the number of items to consider, e.g., the number of tokens in a sequence or the number of pixel patches in an image. $d_k$ is the dimension of a key and query, and $D_v$ is the dimension of a value vector. Queries, keys and values are acquired via a linear projection of the input at each attention layer. For self-attention, they share the same input:

$$Q = E_X M_Q; K = E_X M_K; V = E_X M_V \tag{8}$$

where $M_Q \in \mathcal{R}^{|E_X| \times d_k}$, $M_K \in \mathcal{R}^{|E_X| \times d_k}$ and $M_V \in \mathcal{R}^{|E_X| \times d_v}$ are the linear projection. $E_X \in \mathcal{R}^{N \times |E_X|}$ is the embedding of input $X$, which is jointly represented by its content embedding, $C_X$, and its positional encoding, $P_X$.

$$E_X = C_X \oplus P_X \tag{9}$$

where $\oplus$ can be either concatenation or element-wise addition. Previous work has investigated different combinations and decomposition of positional encoding and content embedding [13]. While concatenation and addition provide comparable results, the lack of positional encoding, $P_X$, will cause a significant drop in accuracy [38, 3, 20]. In this paper, we investigate methods for realizing $P_X$. Note that for all the models except DETR, $P_X$ joins the content embedding as the input to the first layer. For DETR, $P_X$ is added to the input of every Transformer encoder layer, i.e., the activation of the previous Transformer layer.

## B    Learned Positional Encoding Analysis

Our positional encoding is seeded with Fourier features whose dot product approximates $L_2$ distances—that brings the inductive bias to the model, which then evolves as learning progresses. In this section, we analyze the positional encodings learned from the image generation, object detection and widget Captioning tasks. Note that the following analysis is focused on the output of Equation 2 instead of that of Equation 6. The Fourier features directly represent the position while the MLP is trained to modulate the positional encoding to merge with the content embedding. It is less informative to analyze the MLP output because it neither directly represents the position nor directly participates in dot product attention (Equation 7). In all the image benchmarks, the MLP output will be added to the content embedding and the addition is further processed by the transformation with $M_k$ in Equation 8. In the widget captioning benchmark, the MLP output will be concatenated with the content embedding and then projected by a dense layer to a hidden dimension required by the Transformer, which is further transformed by $M_k$ before dot product attention.

### B.1    PE Analysis for Image Generation Tasks

Figure 5 visualizes the similarity of a given position on a $64 \times 64$ image to the rest of the positions on the image, at the initial stage and the end of the training. The similarity is computed based on the dot product of the positional encoding of each position. The first row, Init, shows the similarity heatmap resulted from the initially seeded Fourier features based on $\gamma = 1.0$. The second row, Trained, shows the similarity from the positional encoding learned after 100K steps when the model converges. As we can see, the positional relationship becomes less concentrated than the initialization, i.e., the "ball" becomes larger. To further understand the impact of having the MLP modulator on the positional encoding, we compare the learned positional encoding with and without the MLP modulator. When there is no MLP modulator (Figure 6), the learned positional encoding is less clean

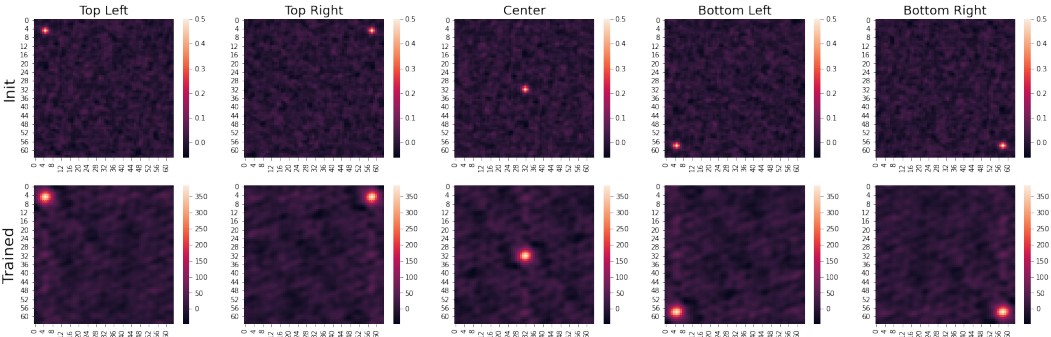

Figure 5: The positional similarity, $r_x \cdot r_y$, of different positions on an image, to the rest of the positions on an image, as learned by Learnable-Fourier+MLP in Reformer. The Fourier features are initialized with weights drawn from a normal distribution: $\gamma = 1.0$. The Top-Left, Top-Right, Center, Button-Left, and Bottom-Right positions are at (4, 4), (4, 57), (31, 31), (57, 4), (57, 57) on the image pixel grid.

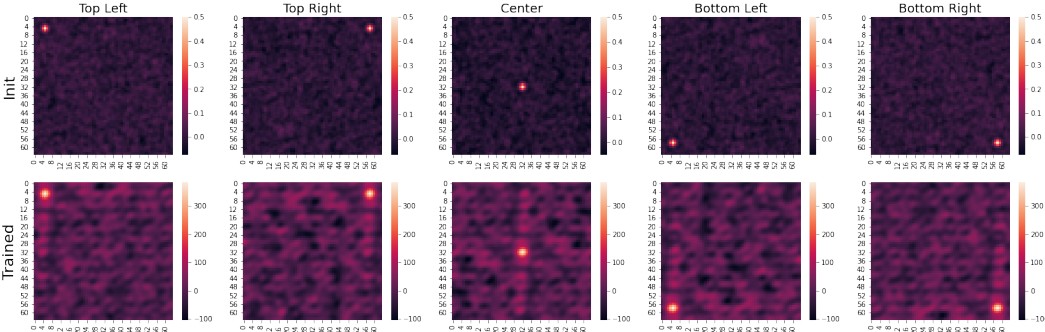

Figure 6: The positional similarity learned by Learnable-Fourier *without using the MLP modulator* in Reformer. The Fourier features are initialized with weights drawn from a normal distribution: $\gamma = 1.0$.

than the one with MLP. We suspect it is because without MLP, the positional encoding needs to directly participate in the addition with the content embedding (Equation 9). As a result, the encoding is not only learning to represent positions but also pressured to work with content embedding. As we show in our experiments, the lack of the MLP modulator results in a decrease in accuracy in this task.

## B.2 PE Analysis for Object Detection Tasks

We visualize the initial and the learned positional relationships of each method in DETR for the object detection task (see Figures 7-10). Similar to the previous analysis, we analyze several representative positions on the $42 \times 42$ grid in DETR, including the Top-Left (5, 5), Top-Right (5, 38), Center (21, 21), Button-Left (38, 5), and Bottom-Right (38, 38) positions, and the heatmaps show the positional similarity of these positions to the rest positions on the grid.

By comparing the similarity heatmap of its initial and trained embedding weights (Figure 7), we found Embed-2D slowly learns spatial relationships between positions, as closer positions becomes more similar (brighter) in the heatmaps. Because the method concatenates independently embedded dimensions, it favors orthogonal directions like Sine-2D, as shown in Figure 2(a).

The learned positional similarity of MLP (Figure 8) is skewed towards the bottom and the right directions based on the five analyzed positions. The heatmap intensity is based on the dot product of PEs, which is not normalized by their magnitudes like cosine similarities. As a result, the heatmap intensity towards the left and top edges is generally smaller (darker). Note that MLP does not have the shift-invariant property and the pattern of these five positions do not necessarily generalize across the entire grid space.

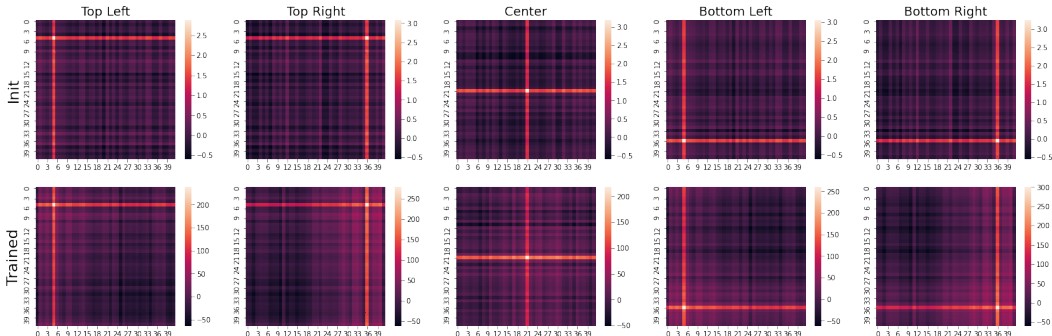

Figure 7: Positional similarity visualization of Embed-2D positional encoding in DETR for object detection.

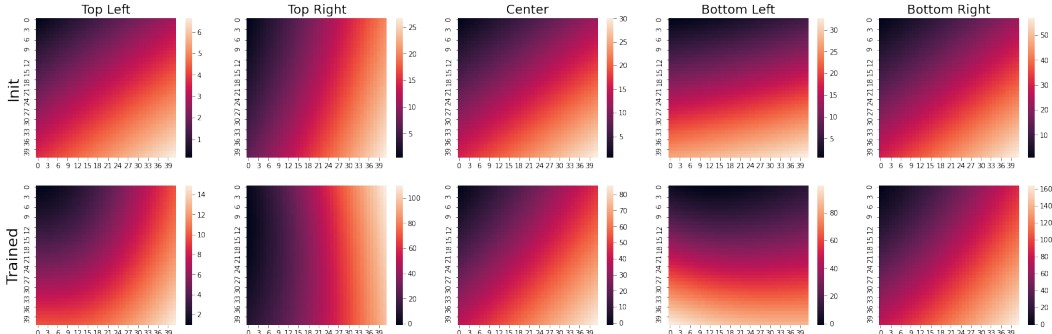

Figure 8: Positional similarity visualization of MLP positional encoding in DETR for object detection.

For Sine-2D, its similarity heatmap obeys the "cross" pattern that we see in Figure 2(a). In DETR, position normalization allows positional encoding to concentrate on the center area of the cross (Figure 9). As a result, the orthogonal bias is much reduced. Finally, we see Learnable-Fourier+MLP was able to mostly maintain ball-shaped similarity pattern throughout the training (Figure 10).

### B.3 PE Analysis for Widget Captioning Tasks

Positional relationships are more complex in the widget captioning task, because each position is defined as a four-coordinate bounding box. We consider point-wise similarity a building block for bounding box similarity as discussed in the paper (Section 4.4). Figure 11 shows the point-wise positional similarity learned by Learned-Fourier+MLP 2/2, which groups four coordinates into two

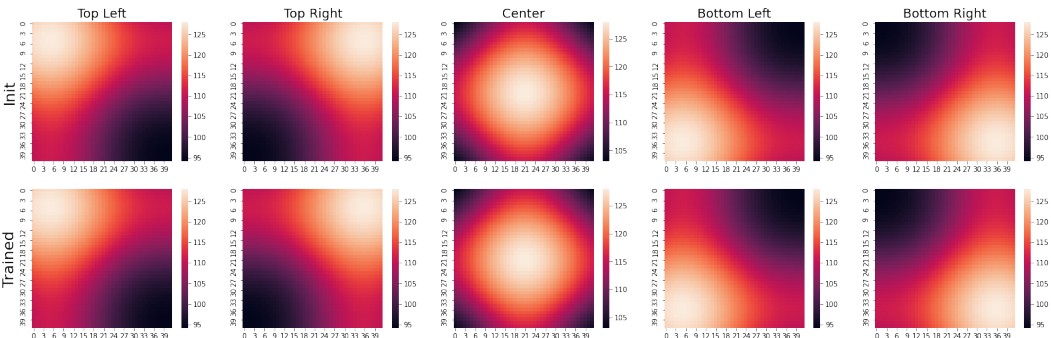

Figure 9: Positional similarity visualization of Sine-2D positional encoding in DETR for object detection. The heatmap of the initial similarity and the "trained" similarity are the same because this method is parameter free.

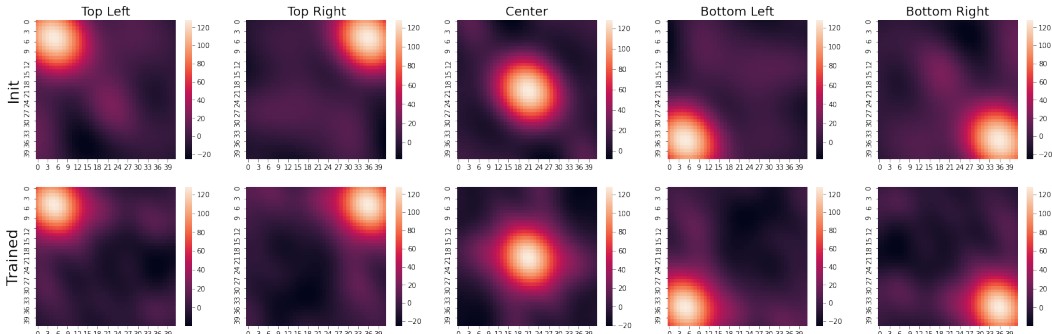

Figure 10: Positional similarity visualization of Learnable-Fourier+MLP in DETR for object detection. The Fourier features are initialized with weights drawn from a normal distribution: $\gamma = 1.0$.

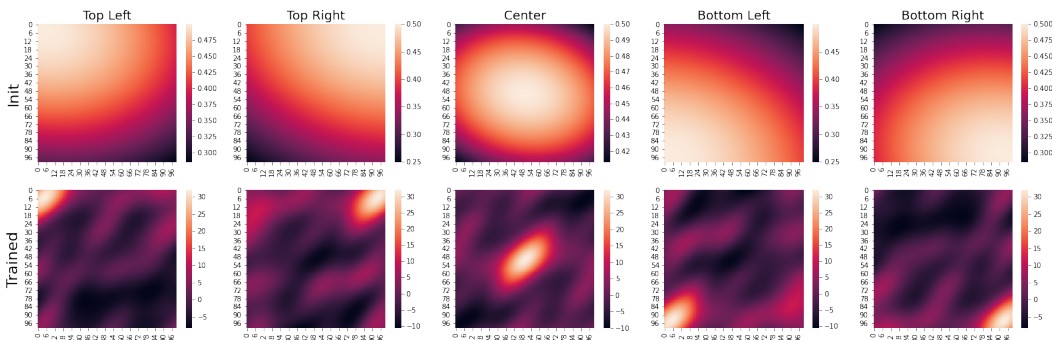

Figure 11: The positional similarity of a UI screen, learned by Learnable-Fourier+MLP-2/2 for widget captioning. Note that in this task, each position is defined as a 4-coordinate bounding box. The heatmap only visualizes the point-wise similarity. The Fourier features are initialized with weights drawn from a normal distribution: $\gamma = 100.0$.

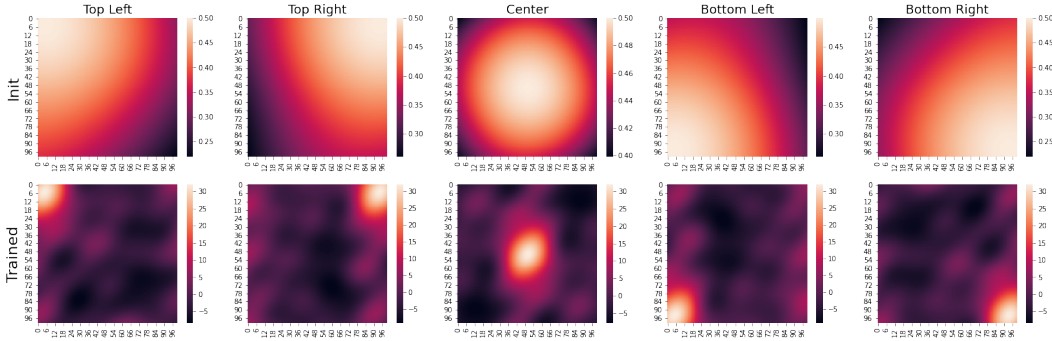

Figure 12: The positional similarity of a UI screen, learned by Learnable-Fourier+MLP-2/2 with the KL loss (Equation 10 and 11) for widget captioning. The Fourier features are initialized with weights drawn from a normal distribution: $\gamma = 100.0$.

groups to represent the top-left corner and the right-bottom corner positions of a bounding box. In this task, we see a more spread positional relationship than that of the image generation task, because we seed the Fourier features with $\gamma = 100$ in this task. We observed that the positional relation becomes more concentrated over the course of the training than that of the initial encodings. We also see the positional relation distribution becomes more skewed (towards the anti-diagonal direction). To understand whether maintain the symmetry of the distribution would help on accuracy, we conduct additional experiments by applying a regularizer to the Fourier weights $W_r$ as the follow.

$$\mathcal{L}_{KL} = -\frac{1}{2}(1 - \log \bar{\sigma}^2 + \log \sigma^2 - \frac{\sigma^2 + \mu^2}{\bar{\sigma}^2}) \tag{10}$$

where $\mu$ and $\sigma^2$ are the mean and variance of $W_r$. $\bar{\sigma}^2$ is the target variance that is also learnable, which is initialized as $\gamma^{-2}$. The KL loss ensures $W_r$ to obey a Gaussian distribution centered at 0 thus maintains the symmetry of positional relationships along all the directions. When training the model, the regularizer loss $\mathcal{L}_{KL}$ is added to the overall loss for optimization.

$$\mathcal{L}_{total} = \mathcal{L}_{model} + \alpha \mathcal{L}_{KL} \tag{11}$$

In this experiment, we use $\alpha = 1$. The resulted positional encoding is shown in Figure 12. As we can see, the symmetry of the positional relation distribution is better maintained with the KL loss, and the distributions of initial and learned $W_r$ for without and with the KL loss are shown in Figure 13. We see a clear improvement of accuracy with the use of this KL loss for Learned-Fourier+MLP 2/2. However, using the KL loss does not seem to impact image-based tasks much, e.g., image generation and object detection tasks. We suspect that as shown in Figure 5, the symmetry of positional relation distribution is naturally maintained even without using the KL loss. Thus KL loss is less useful in such cases.

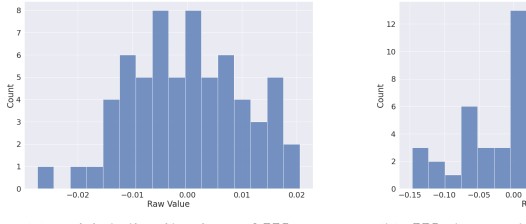
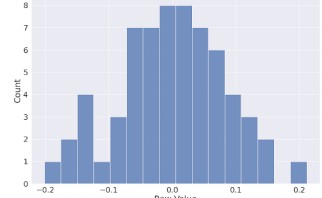
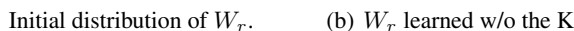

(a) Initial distribution of $W_r$.   (b) $W_r$ learned w/o the KL loss.   (c) $W_r$ learned with the KL loss.

Figure 13: The distribution of $W_r$.

## C   Additional Ablation Studies

It is possible to extend traditional sinusoidal positional encoding (Equation 1) for the multi-dimensional positions by using multi-dimensional frequencies, instead of using the concatenation of independently encoded spatial dimensions. For 2D positions on an image, we can linearly combine the vertical and horizontal positions using constant frequencies that are manually determined. In this ablation, we adapt the original Transformer sinusoidal frequencies for each dimension. Specifically, for a 2D position $(x, y)$, the multi-dimensional sinusoidal PE, referred as *Transformer MD-Sine*, is the follow, where $D$ is the dimension of the PE and $0 \leq d \leq \frac{D}{2}$.

$$PE(p, 2d) = \sin\left(\frac{x}{10000^{2d/D}} + \frac{y}{5000^{2d/D}}\right); PE(p, 2d+1) = \cos\left(\frac{x}{10000^{2d/D}} + \frac{y}{5000^{2d/D}}\right)$$

As shown in Figure 14, Transformer MD-Sine performs poorly in the Reformer Imagenet64 task. Adding MLP to Transformer MD-Since improves its performance, but it still does not perform as good as Learnable Fourier. Although it is possible to find better constant frequencies for linearly combining these dimensions, it can be effort consuming to manually tune these frequencies to perform optimally.

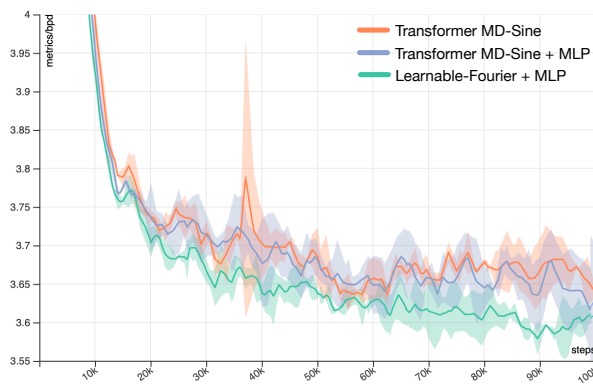

Figure 14: Bits per dim (bpd) w.r.t. training steps on the image generation task with Reformer. The ablation compares learnable Fourier features with multi-dimensional sinusoidal PE based on Transformer frequencies. The plot shows the mean and 95% confidence interval based on 3 repeats of experiments for each method.

Table 5: The performance of Sine-4D when it is enhanced by an MLP for the widget captioning task.

| Method | BLEU-1 | BLEU-2 | ROUGE | CIDEr | METOER | SPICE |
|---|---|---|---|---|---|---|
| Sine-4D | 44.9 | 31.9 | 43.9 | 94.9 | 31.0 | 16.7 |
| Sine-4D+MLP-1/4 | 45.3 | **32.4** | 45.0 | 97.6 | 31.9 | 16.9 |
| Sine-4D+MLP-2/2 | **45.4** | 32.1 | **45.2** | **98.1** | **32.0** | 17.3 |
| Sine-4D+MLP-4/1 | 45.3 | 32.3 | 44.8 | 97.5 | 31.9 | **17.7** |

In contrast, our approach with learnable Fourier features lets the model learn these frequencies that are appropriate for the task.

We found MLP is often beneficial when it is added to an existing positional encoding such as sinusoidal or embedding based methods. For example, the overall accuracy of Sine-4D is improved when an MLP is used with it for the widget captioning task (Table 5). For certain tasks, a dense transform or even simpler scaling over Fourier features (Equation 2) can lead to good results, e.g., the object detection task. Yet, using an MLP seems to consistently offer good results across tasks.

Finally, we compare the performance our positional encoding when $W_r$ is initialized from a different distribution. In this ablation, we initialize $W_r$ by drawing from a uniform distribution in the range of $[0, 1]$ in comparison with drawing from a normal distribution (Equation 4). In the object detection task (Table 6), initializing $W_r$ from the uniform distribution performs worse than from a normal distribution. When the learnable Fourier features are enhanced by the MLP layers, the performance of using both initialization distributions are improved and reach a similar level of performance, although drawing from the normal distribution still has a slight advantage. By examining the learned Fourier features from uniform initialization, we found the positional relationships, as visualized by the heatmaps, has become more "round" or towards a ball shape after learning than those at the initialization (Figure 15), which indicates that the model is more inclined to $L_2$ distances between positions.

Table 6: Performance for initializing $W_r$ with different distributions, and with and without MLP.

| Configuration | $AP$ | $AP_{50}$ | $AP_{75}$ | $AP_{small}$ | $AP_{medium}$ | $AP_{large}$ |
|---|---|---|---|---|---|---|
| Uniform [0, 1] | 38.3 | 59.4 | 40.0 | 17.7 | 41.9 | 56.9 |
| Uniform [0, 1] & MLP | 40.0 | 60.5 | 42.0 | 18.4 | **43.5** | 58.9 |
| $\mathcal{N}(0, 4^{-2})$ | 39.1 | 60.0 | 40.9 | 18.1 | 42.5 | 58.0 |
| $\mathcal{N}(0, 4^{-2})$ & MLP | **40.2** | **60.7** | **42.4** | **20.0** | 43.3 | **59.0** |

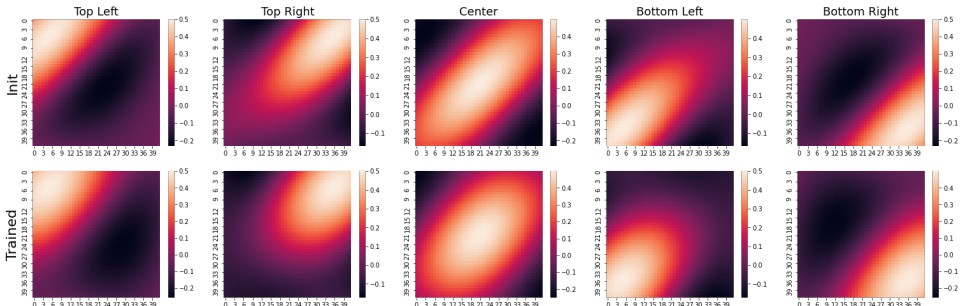

Figure 15: The initial and learned positional relationships of Fourier features when $W_r$ is initialized by drawing from a uniform distribution $[0, 1]$. The Top-Left, Top-Right, Center, Button-Left, and Bottom-Right positions are at (5, 5), (5, 38), (21, 21), (38, 5), (38, 38) in the $42 \times 42$ grid. The Fourier features are initialized with weights drawn from a normal distribution: $\gamma = 4.0$.

## D    Hyperparameters & Parameter Sizes

Table 7: The model parameter sizes of Reformer [16] with different positional encoding methods.

| Method | Reformer Model Parameter Size |
|---|---|
| Embed-1D | 73.2M |
| Embed-2D | 60.7M |
| Sine-1D | 60.6M |
| Sine-2D | 60.6M |
| Transformer MD-Sine | 60.6M |
| Transformer MD-Sine + MLP | 60.7M |
| MLP | 60.6M |
| Learnable-Fourier | 60.6M |
| Learnable-Fourier + MLP | 60.7M |

Table 8: The model parameter sizes of DETR [3] with different positional encoding methods.

| Method | DETR Model Parameter Size |
|---|---|
| Embed-2D | 41.6M |
| Sine-2D | 41.6M |
| MLP | 41.6M |
| Learnable-Fourier + MLP | 41.6M |

For Reformer experiments, each model is based on the Reformer model for the Imagenet64 task [16]. The number of parameters for each Reformer model is summarized in Table 7. We here focus on the positional encoding part of the model that is where each variant differs. Our positional encoding, Learnable-Fourier+MLP, uses roughly the same number of trainable parameters as Embed-2D, the benchmark method used in the original Reformer. All the Fourier-based methods used $|F| = 768$, $|H| = 32$, $D = 768$ and $\gamma = 1.0$. For the MLP, we used LayerNorm before each dense projection, $W_1$ and $W_2$ (see Algorithm 1). We set $G = 1$ because vertical and horizontal positions need to be mapped jointly to model the inductive bias of $L_2$ distances on an image. Embed-1D uses significantly more parameters because it needs to assign an embedding vector for each position in a flattened image. Sine-1D and Sine-2D are parameter-free encoding, thus use the least parameters.

The parameter sizes for each DETR model [3] are shown in Table 8. All the variants of DETR roughly uses the same number of trainable parameters. We used $\gamma = 1.0$ for Learnable-Fourier + MLP in Section 4.2. The MLP uses a dense layer $2 \times 256$ with GeLU as activation.

For UI widget Captioning experiments, the number of parameters of each model variant is shown in Table 9. The model architecture that is shared by each model variant is summarized in the paper and detailed in the previous paper [20]. For Fourier-based methods, we used $|F| = 128, 64, 32,$

Table 9: The model parameter sizes of the widget captioning model [20] with different positional encoding methods.

| Method | Widget Captioning Model Parameter Size |
|--------|---------------------------------------|
| SOTA [20] | 5.11M |
| Embed-4D | 5.11M |
| MLP | 5.08M |
| Sine-4D | 5.07M |
| Sine-4D+MLP-1/4 | 5.07M |
| Sine-4D+MLP-2/2 | 5.07M |
| Sine-4D+MLP-4/1 | 5.08M |
| Learnable-Fourier-2/2 | 5.07M |
| Fixed-Fourier+MLP-1/4 | 5.10M |
| Fixed-Fourier+MLP-2/2 | 5.08M |
| Fixed-Fourier+MLP-4/1 | 5.07M |
| Learnable-Fourier+MLP-1/4 | 5.11M |
| Learnable-Fourier+MLP-2/2 | 5.07M |
| Learnable-Fourier+MLP-4/1 | 5.07M |

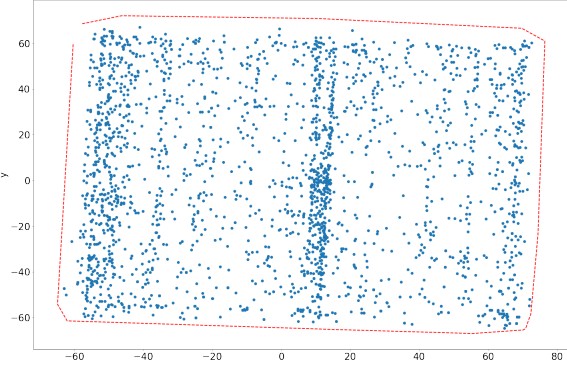

Figure 16: The unseen positions in the test set within the convex hull of all the positions in the training set of the widget captioning dataset.

$G = 1, 2, 4$ for position grouping variants: 1/4, 2/2 and 4/1, respectively. We used $\gamma = 100$ for initializing $W_r$ for all the Fourier-based methods. We used a dropout of 20% after the non-linear activation in the MLP modulator.

For computational complexity, embedding-based approaches generally require less computation than others. For the Reformer experiments, Embed2D, trains at 1.86 steps/second, Sine2D in contrast trains at 1.81 steps/second, and our Learnable-Fourier+MLP is slower and trains at 1.22 steps/second. For the experiments with the object detection and widget captioning tasks, the impact of using different PE methods on runtime is negligible because the rest computation in the training is more dominant, e.g., Hungraian matching for computing the minimal loss in DETR for object detection.

## E  Unseen Position Distribution in the Widget Captioning Dataset

One advantage of the proposed positional encoding method is to generalize to unseen positions. Our experiments with object detection include unseen positions that require "extrapolation". For the widget captioning task, we found 2685 of the 2692 unseen positions in the test dataset are inside the convex hull of all the training positions. To show the distribution of the unseen positions in the test set, we map all these unseen positions to 2D with PCA (see Figure 16). We plot the 2D convex hull of all the positions in the training set, which is also mapped via PCA, as the red dashed line. We can see all the unseen positions are within the convex hull of the training positions for this task.