# OpenReview forum: "Learnable Fourier Features for Multi-dimensional Spatial Positional Encoding"
_NeurIPS.cc/2021/Conference — NeurIPS 2021 Poster_

### Official Review · Reviewer_xpKV · 2021-07-12

**Rating:** 6
**Confidence:** 3

**Summary:**

This paper proposes a new form of positional encoding that introduces more inductive bias (based on L2 distance). The method is a combination of learned Fourier features and MLP transform, demonstrating better performance and faster convergence on both vision and language tasks.

**Limitations And Societal Impact:**

The training curves in Fig 3 & 4 seem to be close to each other, other than the trivial embed-2D. Without the precise numbers, it is hard to determine whether the proposed design brings about significant improvement.

The increased computational complexity and runtime are also concerned, but not mentioned in the original paper. I would appreciate it if the authors can also provide these statistics.


**Main Review:**

The methodology is simple and can replace existing positional encoding methods in place. The authors verify the effectiveness and generality of this new positional encoding on a variety of tasks, including machine translation in NLP, discriminative and generative tasks in CV.

**Time Spent Reviewing:**

4 hours

---

> ### Author Response · Authors · 2021-08-10
> **Responses to Reviewer xpKV**
>
> Thank you very much for your feedback. We address you concerns below, and please let us know if you have further questions.
>
> **Re: Accuracy numbers for Fig 3 and 4**
>
> For Figure 3, we report the accuracy numbers based on the average of the three runs for the top three conditions here (bpd, the smaller the better): Learnable-Fourier+MLP 3.58, Learnable-Fourier 3.64 and Embed2D 3.66. For Fig 4, the $AP$ (the larger the better) for the top three conditions are: Learnable-Fourier+MLP 27.5%, MLP 27%, and Normalized-Sine 2D 26.5%. The $AP_{75}$ (the larger the better) for the top three conditions are: Learnable-Fourier+MLP 27.1%, MLP 26.5%, and Normalized-Sine 2D 25.9%. We will report the accuracy numbers in the revision.
>
> **Re: Complexity & runtime**
>
> For computational complexity, embedding-based approaches generally require less computation compared to others. The Reformer’s default PE, Embed2D, trains at 1.86 steps/second. Sine2D in contrast trains at 1.81 steps/second, and our Learnable-Fourier+MLP is slower and trains at 1.22 steps/second. In Object Detection and Widget captioning, the impact of using different PE on runtime is negligible because the rest computation in the training is more dominant (e.g., Hungraian matching for computing the minimal loss in DETR). We will elaborate on these further in the revision.

---

### Official Review · Reviewer_gVFH · 2021-07-15

**Rating:** 5
**Confidence:** 3

**Summary:**

This paper proposes a learnable positional encoding based on Fourier features for capturing not only positions but also complex positional relationships in the vision domain. The experimental results demonstrate the effectiveness of the proposed method.


**Limitations And Societal Impact:**

I think there is no potential negative societal impact.

**Main Review:**

Overall, the effectiveness of the proposed method does not convince by following concerns;
- Baselines; The proposed method is designed for handling images, and the image classification task is one of the major tasks in computer vision. However, the authors do not demonstrate the effectiveness of the method in the clasification task. Besides, there are several related works that consider improving positional encodings based on Vision Transformer; in detail, Graham et al.[1] introduces attention bias, which lines up with Shaw et al.[2], for providing positional information in each self-attention layer, instead of the input layer. On the other hand, Chu et al.[3] suggests conditional position encoding, which depends on input, by using outputs of depth-wise convolution to the patches.
- Marginal improvements; Figure 4. shows applying MLP already outperforms the baselines and the improvement from adding learnable Fourier is marginal. So, I concern that the most improvements are from the increased number of parameters by adding MLP. Moreover, Figure 3 (a) also shows marginal improvements compared to Embed-2D, which is the original positional encoding of Reformer.
- Clarification; The authors claim that complex spatial relationships can be captured by the proposed method, but the explanation and experimental backups are insufficient.

Minor comments
- Unclear notations; e.g. $\phi$ in eq. 5
- missing ref in Checklist

[1]Graham, B., El-Nouby, A., Touvron, H., Stock, P., Joulin, A., Jégou, H., and Douze, M. Levit: a vision transformer in convnet’s clothing for faster inference.arXiv preprint arXiv:2104.01136,3202021

[2]Shaw, P., Uszkoreit, J., and Vaswani, A.  Self-attention with relative position representations.arXiv preprint arXiv:1803.02155, 2018

[3]Chu, X., Tian, Z., Zhang, B., Wang, X., Wei, X., Xia, H., and Shen, C. Conditional positional308encodings for vision transformers.arXiv preprint arXiv:2102.10882, 2021


**Time Spent Reviewing:**

8

---

> ### Author Response · Authors · 2021-08-10
> **Responses to Reviewer gVFH**
>
> Thank you for your feedback. Please let us know if you have further questions.
>
> **Re: Baselines**
> We will include additional experiments to test our PE in image classification tasks. In particular, we have compared our PE with Embed1D (Vision Transformer’s default PE [Dosovitskiy et. al. ICLR 21]) based on the ViT-B/16 model on ImageNet classification. Following the ViT paper’s hyperparameter settings, we trained each model for 90 epochs. Our PE (Learnable Fourier+MLP) achieved better accuracy (74.5% precision@1) on the validation dataset than Embed1D 73.6%. Since ViT-B/16 uses a 14x14 grid, there aren’t as many positions to represent. We speculate that our PE can achieve more accuracy gain for a larger grid.
>
> We already discussed [2] in the paper, and will cite and discuss [1, 3] that you suggested in the revision. We believe our method is sufficiently different from these works. In addition, our method is fully compatible with original Transformer architectures, which can be directly used in these benchmark models (Reformer, DETR, Widget Captioning) by only swapping the PE part.
>
> **Marginal improvements**
> MLP alone isn’t always enough to bring accuracy gain. Please see Table 1 for the widget captioning task. Our experiments show that Learnable Fourier features plus MLP is the best configuration for PE, which shows consistent improvements across tasks.
>
> Please also notice that in several tasks, the number of parameters of our PE is actually the same or even fewer than that of benchmark models (see Table 1-3 in the Appendix), e.g., Reformer and Widget captioning experiments. As a result, we do not think the improvements are due to the parameter size.
>
> **Clarification**
> In the widget captioning task that involves sparse structures, the spatial relations between two rectangular objects on the screen cannot be simply defined as Euclidean distances. Similarity between two bounding boxes can be related to vertical or horizontal alignment or IoU, and factors that are not obvious. Our PE outperformed benchmark methods in this task, which showed that it is able to capture spatial relationships that are more complex than point distances. We will expand our discussion to clarify this point in the revision.
>
> **Minor comments**
> In Eq. 5, $\phi$ denotes a single layer perceptron. We will clarify the notation in the revision.

---

### Official Review · Reviewer_cSb4 · 2021-07-15

**Rating:** 6
**Confidence:** 4

**Summary:**





**Limitations And Societal Impact:**

Using sine/cosine as an activation function may be problematic during optimization.  This should be mentioned.

**Main Review:**

See Table 8 of Dosovitskiy et.al., equipping with different position embeddings does not produce significantly different results -- this is perhaps due to that the training data and used transformer models in Dosovitskiy et.al. is much larger than in this paper. I am wondering whether the proposed PE also performs better the those PE baselines in image recognition. If it is not, does this mean image generation,  object detection, and widget captioning tasks are more sensitive to the positions of the input? If the authors already did the image recognition, the performance should be reported, otherwise, some discussions may be needed.  So I feel that it may be risky to conclude that the proposed position embedding is generally better than existing ones.

Dosovitskiy et.al. AN IMAGE IS WORTH 16X16 WORDS: TRANSFORMERS FOR IMAGE RECOGNITION AT SCALE. ICLR 2021


It seems that the proposed position encoding is different due to 1)being parameterized by learnable sinusoidal functions 2) being extended to a multi-dimensional case. Note the former may need to be discussed the one in Wang et.al. which proposes learnable sinusoidal PEs. Regarding the latter, one can easily know that Eq. 2 is equivalent to the case when you separately learn learnable  Fourier Features for each dimension (vertical or horizontal dimension) and then concatenate them together. Let me know if I was wrong.

Wang et.al. On position embeddings in BERT

With sin or cosine activations, there are some optimization issues regarding the local minimum, see Parascandolo et.al. It would be better if you could mention this if necessary.

Parascandolo et.al. Taming the waves: sine as activation function in deep neural networks

It may be interesting if you could visualize the learned W_r in eq 2.

**Time Spent Reviewing:**

2 hours

---

> ### Author Response · Authors · 2021-08-10
> **Responses to Reviewer cSb4**
>
> Thank you for your insightful feedback and pointing us to the literature.
>
> **Re: Vision Transformer**
> For Table 8 of Dosovitskiy et.al., the experiments were conducted based on ViT-B/16, which uses the ViT-Base model (86M Params) with a patch size of 16 that is pretrained on the JFT dataset (300 million examples). For patch_size=16, there are only 14x14 unique positions on an image. A simple embedding approach (e.g., Embed1D ViT’s default PE) can perform well, because it is not difficult to learn a good representation for 196 positions with such a large model and dataset. We have conducted additional experiments to compare our PE with Embed1D in ViT-B/16 on the ImageNet dataset. After training for 90 epochs, our PE (Learnable Fourier+MLP) achieved 74.5% precision@1 on the validation dataset compared to 73.6% of Embed1D. We will add these experiments in the revision. We speculate that our PE can achieve more accuracy gain for a larger grid. For example, there are 64x64 positions for the Reformer imagenet64 task, and 100x100 positions for the Widget Captioning task.
>
> **Re: Wang et.al. On position embeddings in BERT & Multidimensional case**
> Thanks for pointing us to the recent work by Wang et. al., which is very relevant. We will cite and discuss the work in the revision. Briefly, we focus on 2D or even higher dimensional positions instead of 1D position in BERT. We also applied MLP on top of learnable sinusoidal, which is a crucial component in our method as shown by all our experiments.
>
> Re: Eq. 2, the concatenation is between the cosine and sine components, where each component is computed based on both horizontal and vertical positions jointly and projected by the shared parameters $W_{r}$. Eq. 2 cannot be decomposed into learning each dimension separately. Please let us know if there is any confusion we should clarify.
>
> **Re: Parascandolo et.al.'s work**
> Thanks for pointing us to Parascandolo et.al.’s work, and we will cite the work and mention the potential optimization problems caused by sine/cosine functions in the revision. Although there may be vanishing gradient problems in extreme cases, we do not observe much difficulty in practice when training the proposed positional encodings.
>
> **Re: Visualization for learned $W_r$**
> We will add a visualization of learned $W_r$ in the Appendix.

---

> > ### Comment · Reviewer_cSb4 · 2021-08-11
> > **Thanks for your response**
> >
> > I increased my score to 6.

---

> > > ### Author Response · Authors · 2021-08-11
> > > **Thank you!**
> > >
> > > Thank you again for your feedback!

---

### Official Review · Reviewer_9tzP · 2021-07-15

**Rating:** 9
**Confidence:** 5

**Summary:**

This paper motivates and introduces a way to do positional encoding that involves a multidimensional sinusoidal encoding with trainable weights followed by a MLP. The method is simple to implement and is advocated as working well.
The balance between the theory and the (very impressive) experiments is good.

**Ethical Concerns:**

nothing particular

**Limitations And Societal Impact:**

no relevant societal impact.
limitations of the work are not highlighted by the authors. maybe that could be interesting to know of further research.

**Main Review:**

I must say that I already reviewed this paper for ICML, and that I was the grumpy reviewer for that paper in that occasion, notably pointing out that some claims were not appropriately justified (novelty on multidimensional sinusoidal encoding) and that the MLP, that seems important, was completely ignored during the discussion.

I was happy to review this paper again because I really hoped that the authors would have fixed these issues, since I believed the contribution is good. And it turns out I am now fully satisfied, because I feel that the authors understood they could not claim to fully understand what happens now in their model (due to the MLP) as they were pretending before. The resulting discussion appears more humble but then way more inspiring and actually more impressive to me. I therefore want to thank the authors for taking the time to work so much on the paper again, notably because of me. But now you got me as a big supporter.


I have some comments still:

Introduction
* L28: ", which" reads wrong

Background
* L105-107: Actually, there were very recent works for relative PE with linear complexity that could be referenced here. (your approach is still different indeed)

3. Learnable Fourier Features positional encoding
* I must say that I appreciate how the authors modified their discussion here to explicitly mention that their proposed representation is a generalization of sinusoidal PE
* In algorithm 1, the title is "compute the PE", while there is an "initialization" step to compute this randomized Wr. This is still a bit confusing.  I understand that somehow this Wr variable is static, so that it is initialized if it's the first run, and kept as it is otherwise, but there is probably a better way to write this. Maybe as describing your positional encoding as a module ? `Constructor` you get the init. Then in the `forward`: your computation ?
* Thanks a lot for including this Transformer MD-sine experiment that I was asking for. This looks quite surprising that the MD-sine+mlp behaves so badly. It hence seems that training the frequencies is crucial here.
* I don't understand the difference between MD-sine and Fourier. Is this the actual frequencies that are picked ? Instead of the regular sampling of the spectrum in MD, you sample in a Gaussian way, that's it ?


**Time Spent Reviewing:**

3

---

> ### Author Response · Authors · 2021-08-10
> **Responses to Reviewer 9tzP**
>
> Thank you so much for your continued feedback, which has been extremely valuable for us to improve the work.
>
> We will revise the writing, and the pseudo code to make it clear that the initialization is executed once as you suggested. We assume the recent linear relative PE paper you pointed out is Liutkus et. al. ICML’21. We will cite and discuss the paper in the revision. You are right that the difference between MD-Sine and Fourier is how the frequencies are determined. For Fourier, we sampled frequencies from a Gaussian distribution. We will discuss the limitations and ideas for future work in the revision, e.g., modeling more complex spatial relations or investigating concatenation versus addition between PE and content embedding.

---

### Official Review · Reviewer_SAzJ · 2021-07-16

**Rating:** 7
**Confidence:** 4

**Summary:**

Modern architectures like transformers relax how data is presented to a network with respect to spatial (or temporal) relationships by allowing the position to be explicitly encoded as part of the input vector rather than implicitly (e.g. in a lattice or sequence). The first transformer model proposed a hand-crafted fixed distributed position encoding based on sinusoidal waves. More recently models like BERT propose to learn the position encoding directly from the data, but this is expensive. This paper proposes a kind of middle ground: a learnable Fourier feature mapping projected by a small MLP. The inductive bias from the Fourier mapping has the advantage of requiring many fewer learnable parameters than a fully unconstrained data driven approach, and naturally extends beyond positions in a sequence to positions in a lattice/grid, etc (existing approaches are based on concatenated dimension-wise embeddings, so capturing similarity on Lp distances for example isn't _directly_ possible - it can be learned with extra functional layers, but then generalisation becomes more problematic empirically).

**Limitations And Societal Impact:**

Whilst technical limitations are covered fairly well, in my view the potential broader societal impacts are insufficiently addressed. The statement in the Broader Impact section reads like the authors are completely disengaged from the current discussion around potential broader (potentially negative) impact. I'd argue that in 2021, for a paper that has real potential for tangible impact in a number of areas, that this is not acceptable.

At the very least I'd encourage the authors to at least read though some of the guidelines from last year's NeurIPS (e.g. https://medium.com/@GovAI/a-guide-to-writing-the-neurips-impact-statement-4293b723f832 and https://brenthecht.medium.com/suggestions-for-writing-neurips-2020-broader-impacts-statements-121da1b765bf) and give this some more thought.

**Main Review:**

Originality
-----------

This is original to the best of my knowledge (but I'd be first to admit to not having read every paper using/proposing a positional embedding). In terms of related work, it certainly covers the papers I am familiar with. The authors have done a good job (see below) of making it clear how this differs from other works.


Quality
-------

Overall, my impression is that what is proposed is a sound idea (it's certainly intuitive enough), and the experimental results validate it in a number of different scenarios. Obviously it would be nice to see how well it works in a large scale language modelling task, but given the resources required it is understandable this might not be achievable.

The authors appear to have done a solid job at evaluating their proposed approach, and have performed multiple repeats where possible, although elicitation of this could be improved (see below).

Clarity
-------
_Is the submission clearly written? Is it well organized? (If not, please make constructive suggestions for improving its clarity.) Does it adequately inform the reader? (Note that a superbly written paper provides enough information for an expert reader to reproduce its results.)_

The authors should be commended for the clarity of the presentation in the early part of the paper (sections 1 & 2). They clearly explained the problem they are trying to address and have given clear (often visual) examples of the shortcomings of the existing approaches. The writing is also self contained - I did not need to refer back to other papers to understand what the point being made is.

Section 3 could be improved a bit:

- Eqn 5 (and the discussion about it) and algorithm 1 are a bit disjoint. Where did $W_p$ go? Why not just say its a two-layer MLP from the outset? Where did $\phi$ go in the algorithm (it's obvious if you read it in detail, but you can make it easier for the reader)
- Alg 1 line 2 - help the reader and maybe just add a comment to say its a ReLU or change the Max(0,x) to ReLU(x)?
- Fig 3. Can you redraw this properly with error bars (or shaded bands) and remove any smoothing (I could be wrong but it looks like the plots might be showing the EMA rather than raw data; the problem with this is that it can remove fluctuations that could ultimately make you draw very different conclusions about performance of different algorithms)

There are a couple of areas where the writing/presentation could be slightly improved (this is rather nit-picky, but hopefully is helpful):

- Line 137: maybe better to say Fourier feature**s** [...] **are**... **a** Gaussian kernel, ...?
- Line 142: feature**s** **have**
- Line 143: are -> have been
- Line 167: introduce**s**
- Line 173/4: **an** RNN
- Line 246: **a** Transformer?
- Line 263: rest -> other
- Tab 1 caption: accuracy -> performance
- Tab 1, 2: reformat according to NeurIPS style (it really does look better!)
- Line 311: treats -> treat
- Fig 3, 4, 5. Please make label & caption text bigger

Significance
------------

Judging significance is hard, but I can see the potential value in the proposed approach if it works on other problems.

**Time Spent Reviewing:**

4

---

> ### Author Response · Authors · 2021-08-10
> **Responses to Reviewer SAzJ**
>
> Thank you very much for your encouraging feedback. We will revise the paper as you suggested, including better connecting equations to Algorithm 1, Figure 3, and grammar errors as you pointed out.
>
> We apologize for the overly brief statement for Broader Impact in our submission. Thanks for pointing us to the guidelines. As a general technique, our technique empowers attention-based models such as Transformer in handling image and spatial structure tasks. For broader societal impacts, it is important to understand the downstream applications or context into which Transformers are used and constrain them appropriately. We will revise and compose our statement for the Broader Impacts carefully in the revision.

---

### Official Review · Reviewer_TqtX · 2021-07-16

**Rating:** 8
**Confidence:** 4

**Summary:**

This paper proposes a new positional encoding (PE) method for transformers. The key idea of this work is to use the Fourier Features technique, which is able to transform the dot product of two vectors into the L2 distance between two positions. By using this technique, the self-attention on PE in transformers, which is essentially dot-product, can be transformed into L2 distance between two positions. Overall, to my understanding, this idea can achieve the effect of relative PE but avoid the problems of relative PE. I think it is really a good idea.

**Limitations And Societal Impact:**

It seems the authors did not mention the limitations of this work. Please include them.

**Main Review:**

Pros:

+ The paper is well-motivated and easy to understand.
+ To me, the proposed idea, i.e., using the fourier features technique in PE, is very interesting. This idea can solve many problems in previous PE methods such as sinusoidal PE or learnable PE. For example, this PE can naturally work well with the self-attention in transformers, and unlike relative PE, it does not need to change the existing implementation of self-attention.
+ In experiments, the proposed methods also show superior performance than other methods.

Cons:

- The biggest issue of this work is in the experiment section. Although the idea shows some positive results in the included experiments, I think the authors should present more results.

For example, how is the performance comparison with relative PE? I understand by design, the idea is better than relative PE, but the performance comparison should also be reported.

Also, on object detection, why did you only show the results in the first 100K steps? I think the final performance should be reported.

How is the performance of the ImageNet classification? The ImageNet classification is one of the most popular tasks, and I think the performance on it is important.


**Time Spent Reviewing:**

1.5 hours

---

> ### Author Response · Authors · 2021-08-10
> **Responses to Reviewer TqtX**
>
> Thank you so much for your encouraging remarks and valuable comments.
>
> **Re: Relative PE**
> Relative PE directly represents pairwise positional relation between query and key, which is typically more expensive and requires some significant change to how dot-product attention is calculated [Shaw et al. NAACL 2018, TransformerXL]. For example, we cannot simply swap the positional encoding in Reformer with Relative PE. Nevertheless, we will compare with Relative PE in some of our tasks.
>
> **Re: Object Detection**
> We intended to show the convergence speed enabled by different PEs in the paper, and we will complete the entire training (300 epochs) and report the final performance in the revision.
>
> **Re: ImageNet Classification**
> We have conducted additional experiments to compare our PE with Embed1D (Vision Transformer’s default PE [Dosovitskiy et. al. ICLR 21]) based on the ViT-B/16 model on the ImageNet dataset. Following the ViT paper’s hyperparameter settings, we trained each model for 90 epochs. We found our PE (Learnable Fourier+MLP) achieved better accuracy (precision@1) on the validation dataset: our PE: 74.5% versus Embed1D: 73.6%. Since ViT-B/16 uses a 14x14 grid, there aren’t as many positions to represent. We speculate that our PE can achieve more accuracy gain for a larger grid. We will add these experiments in the revision.
>
> **Re: Limitation**
> We will add a discussion about the limitation of the work. For example, our work investigated PE when it is combined with content embedding via addition. It would be interesting to investigate how our PE performs when PE and content embedding are concatenated.

---

> > ### Comment · Reviewer_TqtX · 2021-09-03
> > **After rebuttal**
> >
> > I have read the rebuttal and still think the paper present an interesting idea. Thus, I retain my score.

---

### Decision · Program_Chairs · 2021-09-28

**Decision:**

Accept (Poster)

**Comment:**

The majority of reviewers agreed that this paper presents an interesting idea related to a relevant application.

**Consistency Experiment:**

NeurIPS has a long history of experimentation. In 2014, NeurIPS ran an experiment in which 10% of submissions were reviewed by two independent committees to quantify the randomness in the review process. This year, we repeated a variant of this experiment to see how the quality of the review process has changed over time.  This paper was part of the experiment and was therefore assigned to two committees (consisting of reviewers, an Area Chair, and a Senior Area Chair) that reached independent decisions.  If both committees made the same recommendation, this recommendation was followed. If a single committee recommended acceptance, the paper was accepted (with the exception of a few cases in which the other committee identified what we considered a fatal flaw, e.g., an error in a key result).

Both committees reached the same decision: **Accept (Poster)**

The other committee assigned to the paper recommended **Accept (Poster)**.  You can find the other set of reviews, along with any follow up discussion with the authors here:
https://openreview.net/forum?id=YXjhRGvqfFN